# A Novel Autoencoder with Dynamic Feature Enhanced Factor for Fault Diagnosis of Wind Turbine

**Xiaoyin Nie** [1,2,†] 🆔, **Shaoguang Liu** [1,†] 🆔 **and Gang Xie** [1,*] 🆔

1   Shanxi Key Laboratory of Advanced Control and Equipment Intelligence, School of Electronic and Information Engineering, Taiyuan University of Science and Technology, Taiyuan 030024, China; niexiaoyin900113@stu.tyust.edu.cn (X.N.); lsgfly@stu.tyust.edu.cn (S.L.)
2   Department of Automation, Taiyuan Institute of Technology, Taiyuan 030008, China
*   Correspondence: xiegang@tyust.edu.cn; Tel.: +86-0351-699-8022
†   These authors contributed equally to this work.

**Abstract:** Due to the complicated operating environment and variable operating conditions, wind turbines (WTs) are extremely prone to failure and the frequency of fault increases year by year. Therefore, the solutions of effective condition monitoring and fault diagnosis are urgently demanded. Since the vibration signals contain a lot of health condition information, the fault diagnosis based on vibration signals has received extensive attention and achieved impressive progress. However, in practice, the collected health condition signals are very similar and contain a lot of noise, which makes the fault diagnosis of WTs more challenging. In order to handle this problem, this paper proposes a model called denoising stacked feature enhanced autoencoder with dynamic feature enhanced factor (DSFEAE-DF). Firstly, a feature enhanced autoencoder (FEAE) is constructed through feature enhancement so that the discriminative features can be extracted. Secondly, a feature enhanced factor which is independent of manual judgments is proposed and embedded into the training process. Finally, the DSFEAE-DF, combining one noise adding scheme, stacked FEAEs and dynamic feature enhanced factor, is established. Through experimental comparisons, the superiorities of the proposed DSFEAE-DF are verified.

**Keywords:** fault diagnosis; autoencoder; denoising; feature enhancement; dynamic feature enhanced factor

## 1. Introduction

Nowadays, wind energy is becoming one of the most effective means to alleviate energy shortages and protect the environment, so wind turbines (WTs) are widely applied. However, due to the harsh working environment and variable working conditions, WTs are extremely susceptible to failure, resulting in unexpected shutdowns and additional maintenance costs. For example, the shutdown frequency of a wind farm investigated by Caithness Windfarm Information Forum increases from 156 times per year (2009–2014) to 176 times per year (2015–2019), which causes huge economic losses to investors. To reduce maintenance costs and avoid unplanned shutdowns, it is urgent to develop an effective condition monitoring and fault diagnosis model to detect weak faults as early as possible [1–3].

In recent years, the condition monitoring and fault diagnosis of WTs have been greatly developed. In summary, the signals and monitoring means currently employed mainly include the following categories—vibration [4–6], acoustic emission [7,8], strain [9], torque, temperature, oil [10], electrical parameters [11,12], supervisory control and data acquisition (SCADA) parameters [1], non-destructive testing, and so forth. Comprehensive consideration of several aspects such as monitorable componssnts, installation intrusion, installation complexity, installation costs, sampling

frequency requirements, and commercialization, vibration monitoring has become the most widely used approach which provides rich data supports to the development of data-based health monitoring and fault diagnosis.

Benefiting from the development of signal processing and machine learning, many fault diagnosis models have been formed through feature extraction and feature classification [13–17]. Among them, deep neural networks (DNNs), which extract effective features from complex monitoring data automatically and construct a high-reliability model, have gradually become a hotspot in fault diagnosis of WTs [18]. Various DNNs, for example, autoencoder (AE) [13,19,20], sparse filter [4,21], deep belief network (DBN), convolutional neural network (CNN) [22,23], recurrent neural network (RNN) [24], have been employed widely for many challenging problems in fault diagnosis. AE, which minimizes the error in reconstructing the input, can adaptively perform feature extraction in an absolutely unsupervised manner with a simple network structure and few parameters [25–27]. Following this line of reasoning, many variants of autoencoders, for example, denoising autoencoder (DAE), contractive autoencoder (CAE), variational autoencoder (VAE), K-sparse autoencoder, locally connected autoencoder, and so forth., have been proposed recently. For example, considering the noise in signals and non-linearity of signals, Jiang et al. [28] utilized a stacked multilevel-DAE to extract more robust and discriminative fault features. Shen et al. [29] proposed a stacked CAE for anti-noise and robust fault diagnosis. Martin et al. [30] adopted a fully unsupervised deep VAE method for some latent fault feature extraction by variational inferences. These studies motivate us to develop a new AE-based fault diagnosis model for WTs. However, AE is a greedy neural network, and its extracted features are usually trivial. Especially for similar faults, the extracted features of AE are not distinctive and lack of meaning, which guides the AE to focus on important features by adding constraints. L1 regularizer, L2 regularizer, L1L2 regularizer, and KL divergence are widely applied while some of their parameters are usually needed manual judgments. Meanwhile, the signals from WTs operating in variable conditions often contain much noise, which forces AE to still hold the capability of extracting more robust features.

To improve the discriminative and robust feature extraction ability of traditional AE, the denoising stacked feature enhanced autoencoder with dynamic feature enhanced factor (DSFEAE-DF) for fault diagnosis of wind turbine is proposed. The main contributions of this paper are summarized into three folds.

(1) A novel feature enhanced autoencoder (FEAE) is proposed. The FEAE, which introduces feature enhancement, can extract more representative and discriminative features from raw signals.
(2) A dynamic feature enhanced factor is proposed in this paper. The dynamic feature enhanced factor, which involves the diversity of features and information amount between feature and input, is smoothly embedded into the training process and calculated without manual judgments.
(3) DSFEAE-DF is proposed for fault diagnosis of WTs, which involves one noise adding scheme, stacked FEAEs and dynamic feature enhanced factor. Compared to the traditional stacked denoising autoencoder (SDAE), the DSFEAE-DF can extract hierarchical discriminative and robust features and therefore DSFEAE-DF has better ability of similar fault diagnosis and noise environment fault diagnosis.

The remainder of this paper is organized as follows. A brief background of AE is described in Section 2. The proposed model, DSFEAE-DF, is detailed in Section 3 and a set of experiments are conducted for the validity of DSFEAE-DF in Section 4. Finally, the conclusions are drawn in Section 5.

## 2. Background

### 2.1. Autoencoder

Autoencoder (AE) [13], as shown in Figure 1a, is a special DNN which can be divided into two parts: encoding network and decoding network. Given a training sample set $\left\{x^i\right\}_{i=1}^{M}$, where $x^i = \left\{x_1^i; x_2^i, \cdots; x_t^i; \cdots; x_n^i\right\} \in \Re^{n \times 1}$ and $M$ is the number of training samples. The encoding network is to extract a hidden feature $f^i$ from the input sample $x^i$, which is described as

$$f^i = g_{en}\left(\mathbf{W}_{en}\mathbf{x}^i + \mathbf{b}_{en}\right) \in \Re^{d\times1}, \tag{1}$$

where $\mathbf{W}_{en} \in \Re^{d\times n}$ is the weight and $\mathbf{b}_{en} \in \Re^{d\times1}$ is the bias. And the decoding network is to map the hidden feature $f^i$ to the reconstruction output $z^i$, which is described as

$$z^i = g_{de}\left(\mathbf{W}_{de}f^i + \mathbf{b}_{de}\right) \in \Re^{n\times1}, \tag{2}$$

where $\mathbf{W}_{de} \in \Re^{n\times d}$ is the weight and $\mathbf{b}_{de} \in \Re^{n\times1}$ is the bias. The $g_{en}$ and $g_{de}$ represent the activation functions. The training process of AE is to update the parameters $\{\mathbf{W}_{en}, \mathbf{b}_{en}, \mathbf{W}_{de}, \mathbf{b}_{de}\}$ by minimizing the error between $\mathbf{X} = \{\mathbf{x}^i\}_{i=1}^M$ and $\mathbf{Z} = \{\mathbf{z}^i\}_{i=1}^M$ as follows:

$$L\left(\mathbf{X}, \mathbf{Z}\right) = \frac{1}{M}\sum_{i=1}^M \left\| \mathbf{x}^i - \mathbf{z}^i \right\|^2. \tag{3}$$

*2.2. Denoising*

Due to the complicated environment, the collected signals often contain strong background noise, so the performance of AE with noise needs to be improved. To learn more robust features, noise sample $\tilde{\mathbf{x}}^i$ is constructed by input sample $\mathbf{x}^i$ through adding noise by $q_D$, which is denoted as $\tilde{\mathbf{x}}^i \sim q_D\left(\tilde{\mathbf{x}}^i|\mathbf{x}^i\right)$. Then, through the encoding network and the decoding network, the noise sample $\tilde{\mathbf{x}}^i$ are mapped to the reconstruction output $z^i$. Finally, by minimizing error in Equation (3), robust features are extracted.

**3. Proposed Method**

In this section, the proposed DSFEAE-DF is firstly presented, including FEAE, dynamic feature enhanced factor and the structure of DSFEAE-DF. Then, the fault diagnosis procedures are detailed.

*3.1. Feature Enhancement*

When observing the features extracted by AEs, they are not discriminatively different. That is, AEs greedily extract relatively trivial features to reconstruct input samples.To overcome this shortcoming, one approach guides the AE to focus on important features by adding constraints in the training process via mutual competition and enhancement. In competition and enhancement, neurons in the hidden layer compete for the right to respond to the input samples, then the specialization of neurons increases so that discriminative features can be extracted. Following this idea and inspired by Reference [31], the feature enhancement is proposed as two processes, as described below and detailed in Algorithm 1.

(1) Competition: In the feature vector $f^i = \{f_1^i; f_2^i; \cdots; f_t^i; \cdots; f_d^i\}$, the most competitive $k$ neurons with the largest activation values are selected as the "winner" in competition, while the remaining "loser" are suppressed as 0.
(2) Enhancement: In order to compensate for the energy loss caused by suppressing the "loser" neurons and make the competition among the neurons more obvious, the average "loser" neuron energy $E\left(F_{1,d-k}^i\right)$ is redistributed to the "winner" neurons by energy enhanced factor $\beta$, which achieves the enhancement. Given a feature enhanced factor $\alpha$, the most competitive $k$ and energy enhanced factor $\beta$ can be denoted as below.

$$k = \lceil \alpha \cdot d \rceil \tag{4}$$

$$\beta = {}^1/_\alpha. \tag{5}$$

---

**Algorithm 1** function $feature\_enhancement(f^i)$

---

**Input:** feature vector $f^i = \{f^i_1; f^i_2; \cdots; f^i_t; \cdots; f^i_d\}$, feature enhanced factor $\alpha$, most competitive $k = [\alpha \cdot d]$, energy enhanced factor $\beta = 1/\alpha$

**Output:** $h^i = \{h^i_1; h^i_2; \cdots; h^i_t; \cdots; h^i_d\}$

1: Let $f^i = \{f^i_1; f^i_2; \cdots; f^i_t; \cdots; f^i_d\}$ in ascending order into $f'^i = \{f'^i_1; f'^i_2; \cdots; f'^i_{t'} \cdots; f'^i_d\}$, and record index list $[index]$

2: Average "loser" neuron energy $E\left(F^i_{1,d-k}\right) = \frac{1}{d-k} \sum\limits_{j=1}^{d-k} f'^i_j$

3: **for** $j = d - k + 1; j \leq d; j + + $ **do**

4:　　$h'^i_j = f'^i_j + \beta \cdot E(F^i_{1,d-k})$

5: **end for**

6: **for** $j = 1; j \leq d - k; j + + $ **do**

7:　　$h'^i_j = 0$

8: **end for**

9: Sort $h'^i = \{h'^i_1; h'^i_2; \cdots; h'^i_t; \cdots; h'^i_d\}$ back to $h^i = \{h^i_1; h^i_2; \cdots; h^i_t; \cdots; h^i_d\}$ according to the $[index]$

　　**return** enhanced feature vector $h^i = \{h^i_1; h^i_2; \cdots; h^i_t; \cdots; h^i_d\}$.

---

### 3.2. Feature Enhanced Autoencoder

According to the descriptions of AE in Section 2.1 and feature enhancement in Section 3.1, feature enhanced autoencoder (FEAE), as shown in Figure 1b, is proposed as Equations (1), (6) and (7), where $f^i$ is denoted as feature and $h^i$ is denoted as enhanced feature. FEAE, as an unsupervised feature extractor, can extract discriminative features through feature enhancement. Similarly, the parameters $\theta = \{W_{en}, b_{en}, W_{de}, b_{de}\}$ can be trained by minimizing the Equation (3).

$$h^i = feature\_enhancemnet\left(f^i\right) \in \Re^{d \times 1} \tag{6}$$

$$z^i = g_{de}\left(W_{de} h^i + b_{de}\right) \in \Re^{n \times 1} \tag{7}$$

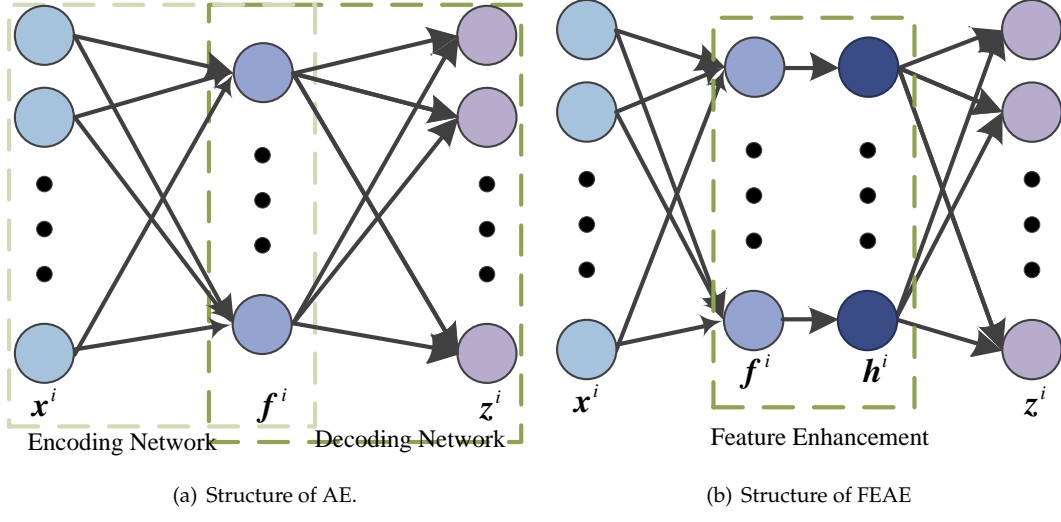

(a) Structure of AE.　　　　　　　　　　　　　　(b) Structure of FEAE

**Figure 1.** Structures of autoencoder (AE) and feature enhanced autoencoder (FEAE).

### 3.3. Dynamic Feature Enhanced Factor

As can be seen in Section 3.1, feature enhanced factor $\alpha$, as a hyperparameter, is a key factor of feature enhancement and represents the proportion of most competitive $k$ neurons in the total neurons. With a large $\alpha$, $k$ is also large so that too many features are enhanced and the significance of feature enhancement decreases. While with a small $\alpha$, $k$ is also small causing few features are enhanced and the remaining features are set to 0 so that the features are lost. In the existing method [31], a stable $\alpha$ is employed by prior knowledge and human judgment, but due to the complexity and diversity of overall $\{f^i\}_{i=1}^{M}$, a stable $\alpha$ cannot be suitable for all features, so a dynamic feature enhanced factor $\alpha^b$ independent on human judgment is proposed as follows.

As for the training process, the batch size is $B$, the current batch is $b$, then the training sample set in current batch $b$ is $X^b = \{x^i\}_{i=1+B\cdot(b-1)}^{B+B\cdot(b-1)}$. Through the encoding network, the feature set $F^b = \{f^i\}_{i=1+B\cdot(b-1)}^{B+B\cdot(b-1)}$ can be obtained. Then, the similarity between the $f^i$ and the rest features in $F^b$ can be denoted as

$$sim\left(f^i\right) = 1 - \frac{1}{2\left(B-1\right)} \sum_{j=1+B\cdot(b-1),j\neq i}^{B+B\cdot(b-1)} \left[ \left(\frac{\bar{f}^i - \bar{f}^j}{\bar{f}^i + \bar{f}^j}\right)^2 + \left(\frac{v^i - v^j}{v^i + v^j}\right)^2 \right], \tag{8}$$

where $\bar{f}^i = \frac{1}{d}\left(f_1^i + f_2^i + \cdots + f_d^i\right)$ and $v^i = \frac{1}{d}\left[\left(f_1^i - \bar{f}^i\right)^2 + \left(f_2^i - \bar{f}^i\right)^2 + \cdots + \left(f_d^i - \bar{f}^i\right)^2\right]$. The average similarity $sim^b$ of $F^b$ is calculated as

$$sim^b = \frac{1}{B} \sum_{i=1+B\cdot(b-1)}^{B+B\cdot(b-1)} sim\left(f^i\right). \tag{9}$$

Meanwhile, the information amount $I$ between feature $f^i$ and input sample $x^i$ is designed as

$$I = -\log_{10}\left(\frac{d}{n}\right), \tag{10}$$

where $d$ and $n$ are the dimensions of $f^i$ and $x^i$ respectively. Finally, in current batch $b$, the dynamic feature enhanced factor $\alpha^b$ is designed as

$$\alpha^b = \frac{1}{2}\left(\left(1 - sim^b\right) + I\right). \tag{11}$$

Furthermore, the $k^b$, $\beta^b$ can be calculated by Equations (4) and (5).

Interpreting Equations (11), the dynamic feature enhanced factor $\alpha^b$ consists of two terms. The first term $\left(1 - sim^b\right)$ is the average diversity of $F^b$, which can be roughly regarded as the proportion of discriminative features. The second term $I$ represents the information amount carried by the features from the input samples, which prevents too few features that can be enhanced due to the small average diversity of $F^b$ and ensures enough enhanced features to reconstruct the input. As can be seen in the above descriptions, the dynamic feature enhanced factor $\alpha^b$ is proposed reasonably and acquired adaptively without prior knowledge in every training batch.

### 3.4. DSFEAE-DF Model

The proposed denoising stacked feature enhanced autoencoder with dynamic feature enhanced factor (DSFEAE-DF) model is a DNN with one noise adding scheme, multiple feature enhanced layers, and one softmax layer, as shown in Figure 2.

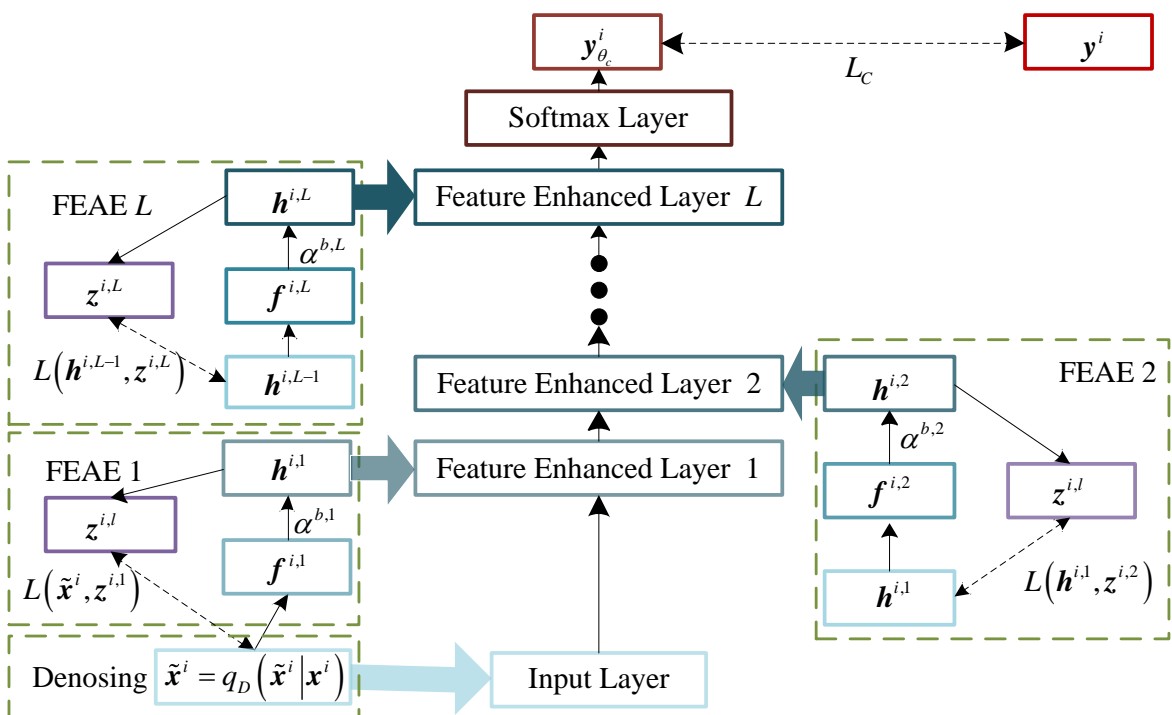

**Figure 2.** Structure of denoising stacked feature enhanced autoencoder with dynamic feature enhanced factor (DSFEAE-DF) model.

The feature enhanced layers are composed of a set of FEAEs to achieve discriminatively and automatically extraction of enhanced features at different layers from the original signals. Assuming that the multiple feature enhanced layers has $L$ FEAEs, $l \in \{1, 2, 3, \ldots, L\}$ represents the $l$-th FEAE. When $l = 1$, the input $x^{i,1}$ of the first FEAE is described as $x^{i,1} = \tilde{x}^i$, which means that the input of DSFEAE-DF is the noise sample $\tilde{x}^i$. Then enhanced features $h^{i,1}$ in first FEAE can be obtained by updating $\theta^1 = \{W_{en}^1, b_{en}^1, W_{de}^1, b_{de}^1\}$. And when $l = 2, 3, \ldots, L$, the input $x^{i,l}$ of the $l$-th FEAE is $h^{i,l-1}$, and update $\theta^l = \{W_{en}^l, b_{en}^l, W_{de}^l, b_{de}^l\}$ to get the enhanced features $h^{i,l}$.

The softmax layer, whose input is $h^{i,L}$, is employed to make the prediction $y_{\theta_c}^i \in \Re^{C \times 1}$ of the input sample $x^i$. Supposing that the label of $x^i$ is $y^i$, the discrepancy between $y_{\theta_c}^i$ and $y^i$, computed by the crossentropy loss function $L_c$ in Equation (12), reach the minimum through updating the weight $\theta_c$ of softmax layer.

$$L_c = -\frac{1}{M} \sum_{i=1}^{M} (y^i)^T \log y_{\theta_c}^i, \tag{12}$$

where $y^i \in \Re^{C \times 1}$ is one-hot form of label $y^i$.

Consequently, the stacked multiple FEAE layers can obtain hierarchical non-linear features, where enhanced features of lower layers are extracted in lower layers and enhanced features of higher layers are extracted in the higher layers. Meanwhile, the dynamic feature enhanced factor $\alpha^{b,l}$ of the $l$-th FEAE in every batch can be smoothly embedded into the model and adaptively calculated by Equation (11) during the training process so that enhanced features are extracted automatically without any human judgment. Furthermore, the input of the first FEAE is the noise sample $\tilde{x}^i$ by one noise adding scheme, which corresponds to the denoising described in Section 2.2.

### 3.5. DSFEAE-DF for Fault Diagnosis

The fault diagnosis procedures based on the DSFEAE-DF model includes two phases: training phase and testing phase. During the training phase, the different health condition vibration signals are collected, segmented, normalized, and put into the established DSFEAE-DF model. Next, complete

the training of the model and obtain the trained DSFEAE-DF. The detailed training process of the training phase is shown in Algorithm 2. In the testing phase, new health condition signals are acquired, segmented, normalized, and fed into the trained model, then the diagnosis result can be obtained.

---

**Algorithm 2** Training process of DSFEAE-DF

---

**Input:** Dataset $\{x^i, y^i\}_{i=1}^M$; Model DSFEAE-DF; Number of epochs $E$; Batch size $B$; Noise adding
    probability $q_D$

**Output:** $\{\theta^l\}_{l=1}^L$ and $\theta_c$

 1: #Unsupervised training layer by layer

 2: Corrupt $x^i$ into $\tilde{x}^i$ with $q_D$, initialize parameters $\{\theta^l\}_{l=1}^L$.

 3: **for** $l = 1 \rightarrow L$ **do**

 4:     **if** $L = 1$ **then**

 5:        $x^{i,l} = \tilde{x}^i$

 6:     **else**

 7:        $x^{i,l} = h^{i,l-1}$

 8:     **end if**

 9:     **for** $e = 1 \rightarrow E$ **do**

10:        **for** $b = 1 \rightarrow M/B$ **do**

11:           $f^{i,l} = g_{en}\left(W_{en}^l x^{i,l} + b_{en}^l\right)$

12:           $\alpha^{b,l}$, $k^{b,l}$ and $\beta^{b,l}$ obtained in Equations (11), (4), (5)

13:           $h^{i,l} = feature\_enhancement\left(f^{i,l}\right)$ and $z^{i,l} = g_{de}\left(W_{de}^l h^{i,l} + b_{de}^l\right)$

14:           Update $\theta^l$ by minimizing Equation (3)

15:        **end for**

16:     **end for**

17: **end for**

18: #Supervised fine-tuning

19: **for** $e = 1 \rightarrow E$ **do**

20:     Predict $y_{\theta_c}^i$

21:     Update $\theta_c$ by minimizing Equation (12)

22: **end for**

---

## 4. Experiments and Verification

In this section, since the bearing is a core component of WTs, the dataset of bearing is applied to verify the effectiveness of the proposed method. All experiments are conducted with a computer with AMD A8-5550M APU, Linux OS, and Tensorflow Toolbox. All experiments repeat 10 trails to avoid the one-time occasionality.

### 4.1. Data Description

The bearing vibration signals of Case Western Reserve University (CWRU) [32] are employed in this paper. All signals are obtained from artificially damaged bearings in the motor driving mechanical system shown in Figure 3. The signals under four fault locations (Normal, Ball, Inner race and Outer race) are collected by the acceleration sensors under four different loads with 48 kHz sampling frequency. For each fault location, three fault severities (0.18 mm, 0.36 mm and 0.53 mm) are introduced, respectively. We use these situations to simulate actual bearing faults in WTs.

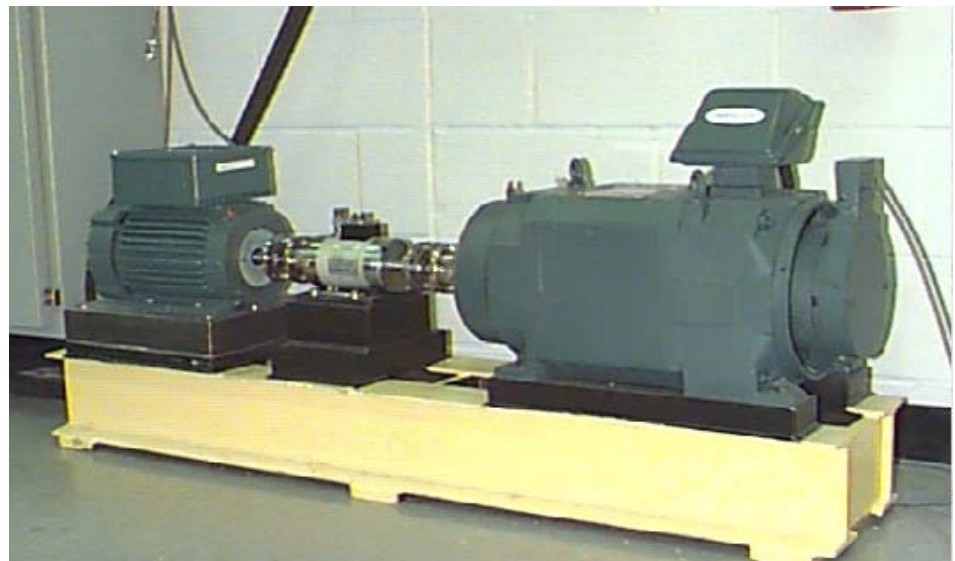

**Figure 3.** Motor driving mechanical system.

The vibration signal under one load and under one fault location with one fault severity contains 240, 000 data points during 5 s sampling time. Through segmenting, 200 segments with a length of 1200 data points are obtained from one vibration signal. Through the same operation for all signals, the dataset is formed and detailed in Table 1. The dataset contains a total of 10 health conditions. For each health condition, there are 800 segments (200 segments under each load) and a segment (also called a sample) contains 1200 data points, which means the dataset containing a total of 8000 samples. Figure 4 provides examples of 10 health condition samples.

**Table 1.** Dataset description.

| Label | Fault Location | Fault Severity (mm) | Load (hp) | Data Point Number of Sample | Sample Number |
|---|---|---|---|---|---|
| 0 | Normal | 0 | 0,1,2,3 | 1200 | 200,200,200,200 |
| 1 | Ball | 0.18 | 0,1,2,3 | 1200 | 200,200,200,200 |
| 2 | Ball | 0.36 | 0,1,2,3 | 1200 | 200,200,200,200 |
| 3 | Ball | 0.53 | 0,1,2,3 | 1200 | 200,200,200,200 |
| 4 | Inner race | 0.18 | 0,1,2,3 | 1200 | 200,200,200,200 |
| 5 | Inner race | 0.36 | 0,1,2,3 | 1200 | 200,200,200,200 |
| 6 | Inner race | 0.53 | 0,1,2,3 | 1200 | 200,200,200,200 |
| 7 | Outer race | 0.18 | 0,1,2,3 | 1200 | 200,200,200,200 |
| 8 | Outer race | 0.36 | 0,1,2,3 | 1200 | 200,200,200,200 |
| 9 | Outer race | 0.53 | 0,1,2,3 | 1200 | 200,200,200,200 |

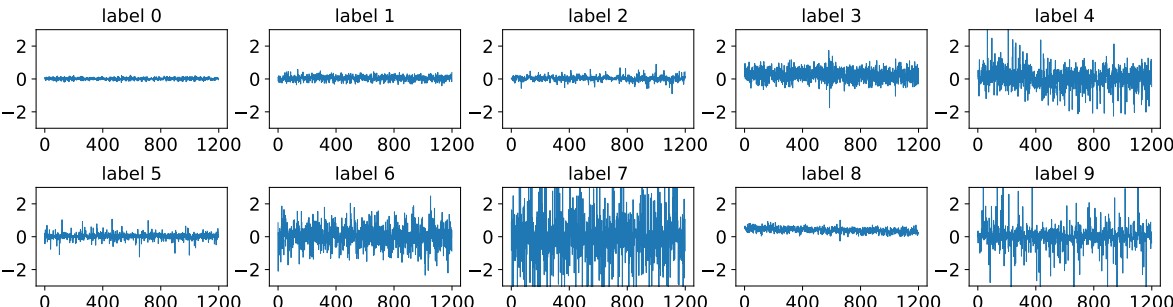

**Figure 4.** Each health condition sample.

### 4.2. Experimental Setup

The parameters of the DSFEAE-DF structure and training process have an impact on testing accuracy. To determine these parameters, exhaustive experiments on the bearing dataset are undertaken

to obtain the optimized parameters and the testing accuracy is set to be the indicator. The parameters of five aspects under consideration are presented as follows.

(1) The structure of DSFEAE-DF. Combining the structure in Reference [33] and testing accuracy, the structure of DSFEAE-DF includes an input layer, three feature enhanced layers, and a softmax layer. The dimension of the input layer is equal to the dimension of the input sample, and the dimensions of the three feature enhanced layers are 600, 400, and 200, respectively. The number of nodes in the softmax layer is 10, which is the number of health conditions.

(2) Activation function. Compared to other nonlinear functions, for example, ReLU, Tanh, LogSig, and SoftSign, the $g_{en}$ and $g_{de}$ utilize the Sigmoid function.

(3) Noise adding probability. The Gaussian noise is adopted in this paper. Combing noise settings in Reference [1] and testing accuracy, the noise adding probability $q_D$ is set as 0.1.

(4) Optimization. Following Reference [34], Adam is adopted for stochastic optimization. The learning rates of the unsupervised training and the supervised fine-tuning are set to 0.01 and 0.0165, respectively. Comprehensive consideration of the degree of optimization and the speed of optimization, the epoch is set to 200 and batch size is set to 100.

(5) Training sample number. Following the setting in Reference [35], random 7200 samples are used for training and the rest are used for testing.

To verify the effectiveness of dynamic feature enhanced factor $\alpha^{b,l}$, we embed 9 values of $\alpha$ into the feature enhancement of the DSFEAE for comparison to DSFEAE-DF in fault diagnosis. Otherwise, in order to verify the performances of the proposed model in similar fault diagnosis and noise environment fault diagnosis, other AE-based models with the same establishment of DSFEAE-DF are employed for comparisons. The details are as follows.

(1) SAE: a stacked AE, a traditional model described in Reference [33].

(2) K-sparse SAE: the stacked AE with K-sparse, where K-sparse proposed in Reference [36] means only competition process but no enhancement process;

(3) SDAE: the stacked denoising AE, a traditional model described in Reference [37];

(4) DSFEAE-0.4: the denoising stacked FEAE with the stable feature enhanced factor 0.4.

*4.3. Effectiveness of Dynamic Feature Enhanced Factor*

In this paper, the consideration of the diversity of corresponding features in a training batch and the information amount between feature and input in the current feature enhanced layer, the dynamic feature enhanced factor $\alpha^{b,l}$ is designed, which is smoothly embedded into the training process. To verify the effectiveness of the dynamic feature enhanced factor, 9 values of $\alpha$ are embedded in the feature enhancement of the DSFEAE. The diagnosis results of manually setting parameters $\alpha$ and dynamic $\alpha^{b,l}$ are shown in Figure 5. It can be seen that as the value increases from 0.1 to 0.4, the testing accuracy continuously increases, while the value increases from 0.4 to 0.9, the testing accuracy continuously decreases. That is because a small value means a small number of feature are enhanced and little information would be contained in enhanced features, which makes it difficult to reconstruct input samples. While too larger value means most of the features are enhanced so that the meaning of feature enhancement decreases. The DSFEAE-DF uses dynamic feature enhanced factor $\alpha^{b,l}$ which can adaptively select features and enhance them. The accuracy of DSFEAE-DF is higher than that of DSFEAE with any stable $\alpha$, and its standard deviation is also smaller than others. These results verify the effectiveness of the dynamic feature enhanced factor.

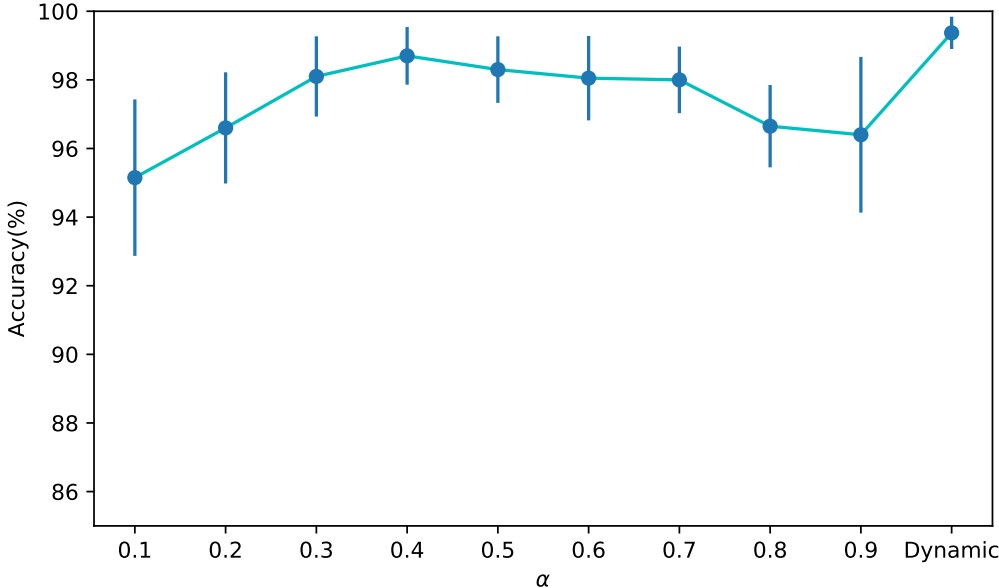

**Figure 5.** Accuracies of DSFEAE with stable $\alpha$ and DSFEAE-DF.

*4.4. Performance of Fault Diagnosis*

In this subsection, the performances of fault diagnosis and similar fault diagnosis are discussed. Table 2 shows the diagnosis results of the five models. The SAE achieves the worst accuracy of 91.30% with a standard deviation of 2.34%. K-sparse SAE, SDAE and DSFEAE-0.4 acquire the middle accuracies, which are 94.72%, 97.14% and 98.70% with standard deviations of 2.07%, 1.89% and 0.84%. It can be seen that DSFEAE-DF obtains the highest accuracy of 99.37% and the lowest standard deviation of 0.47%, which indicates the fault diagnosis performance of DSFEAE-DF is superior and stable.

**Table 2.** Fault diagnosis results

| Accuracy (%) | SNR (dB) | | | | | | | | No Noise |
|---|---|---|---|---|---|---|---|---|---|
| | −4 | −2 | 0 | 2 | 4 | 6 | 8 | 10 | |
| SAE | 46.80 ±3.38 | 59.45 ±3.38 | 69.85 ±3.27 | 80.35 ±2.89 | 83.55 ±3.52 | 84.60 ±3.09 | 87.75 ±3.27 | 90.45 ±2.78 | 91.30 ±2.34 |
| K-sparse SAE | 47.95 ±4.15 | 60.25 ±4.25 | 71.55 ±4.62 | 81.95 ±4.19 | 87.05 ±5.32 | 87.55 ±3.41 | 88.65 ±2.67 | 91.40 ±3.38 | 94.72 ±2.07 |
| SDAE | 52.05 ±3.62 | 69.00 ±4.02 | 78.90 ±3.12 | 86.10 ±4.59 | 90.65 ±2.99 | 93.25 ±1.92 | 94.30 ±2.06 | 95.50 ±1.59 | 97.14 ±1.89 |
| DSFEAE-0.4 | 57.00 ±5.27 | 77.10 ±2.88 | 88.60 ±3.79 | 93.89 ±1.38 | 96.10 ±1.64 | 97.30 ±1.70 | 97.62 ±1.32 | 98.00 ±1.25 | 98.70 ±0.84 |
| DSFEAE-DF | 67.15 ±3.83 | 82.55 ±3.48 | 91.55 ±3.25 | 93.92 ±1.78 | 96.30 ±1.41 | 97.43 ±1.31 | 97.80 ±1.20 | 98.20 ±0.70 | 99.37 ±0.47 |

Further, we explore the performance of similar fault diagnosis, so the confusion matrices of the testing results, which can detail the classification of each health condition, are drawn in Figure 6. According to Figure 4, the samples of the label 4 and label 9 are similar, so the classification of label 4 and label 9 is challenging. In Figure 6, 14.8% of samples of label 9 are misclassified as label 4 in SAE, 11.25% in K-sparse SAE, 10.30% in SDAE, 3.6% in DSFEAE-0.4, but only 1.08% in DSFEAE-DF. Similar situations also occur between label 1 and label 2 and between label 3 and label 4, which confirm the capability of classification of similar faults in DSFEAE-DF. Meanwhile, not only the average accuracy but also the accuracy of each label in DSFEAE-DF are generally higher than with other models. All these results illustrate the superiority of similar faults diagnosis.

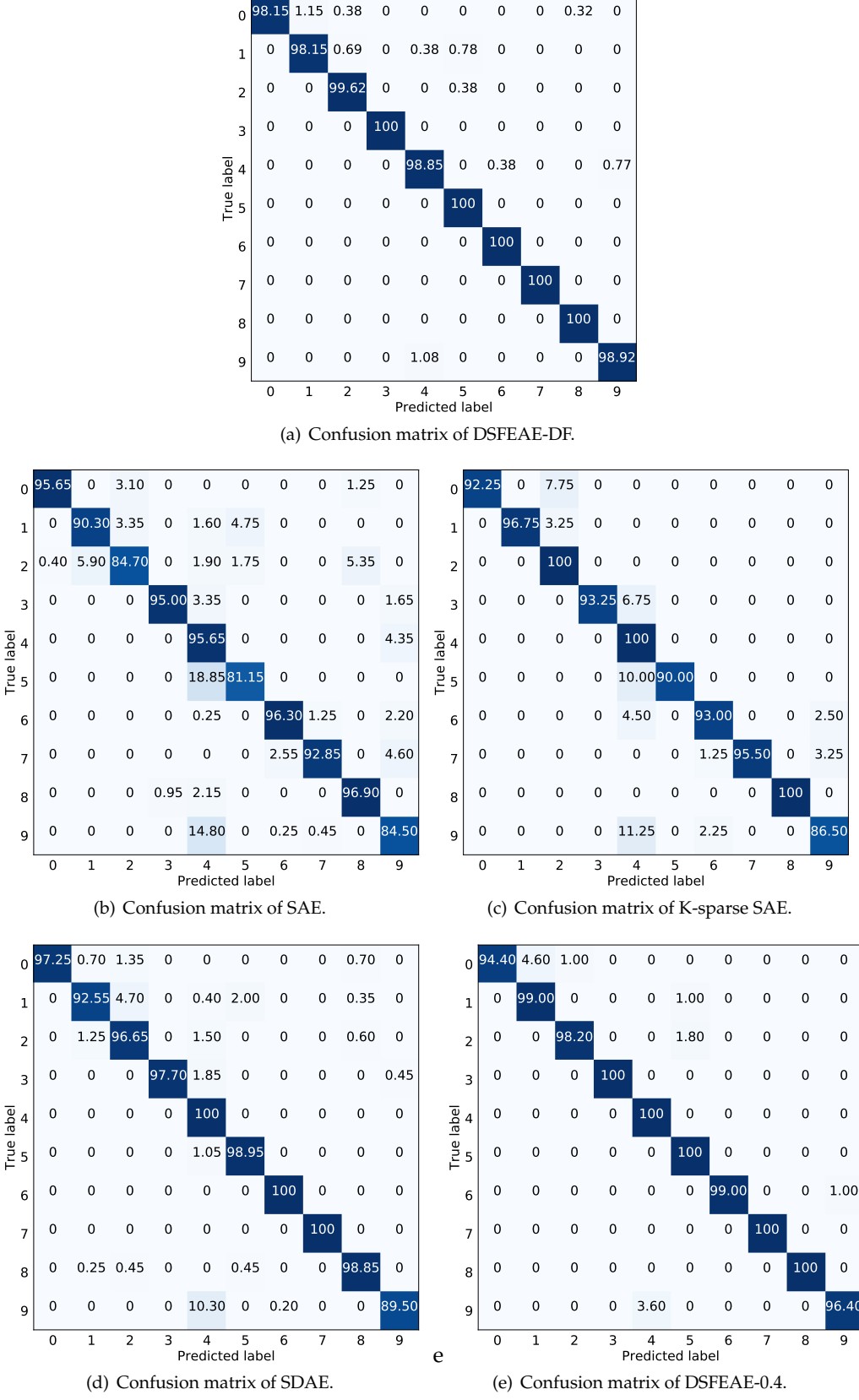

**Figure 6.** Confusion matrices of five models.

In Figure 7, the *t*-distributed stochastic neighbor embedding (*t*-SNE) [38] is used to map the features of last enhanced feature layer to two-dimensional features, thereby displaying the fault

diagnosis results more intuitively. It can be seen that the features of SAE, K-sparse SAE, and SDAE are partially overlapped and the clustering effect is not satisfying. And in DSFEAE-0.4, the clustering effect is good but some similar faults overlapped slightly, which matches the confusion matrix shown in Figure 6e. For the DSFEAE-DF, the features of the same health condition can be clustered together, and the separation between the features of different health conditions is clearer. These can verify the superiority of DSFEAE-DF.

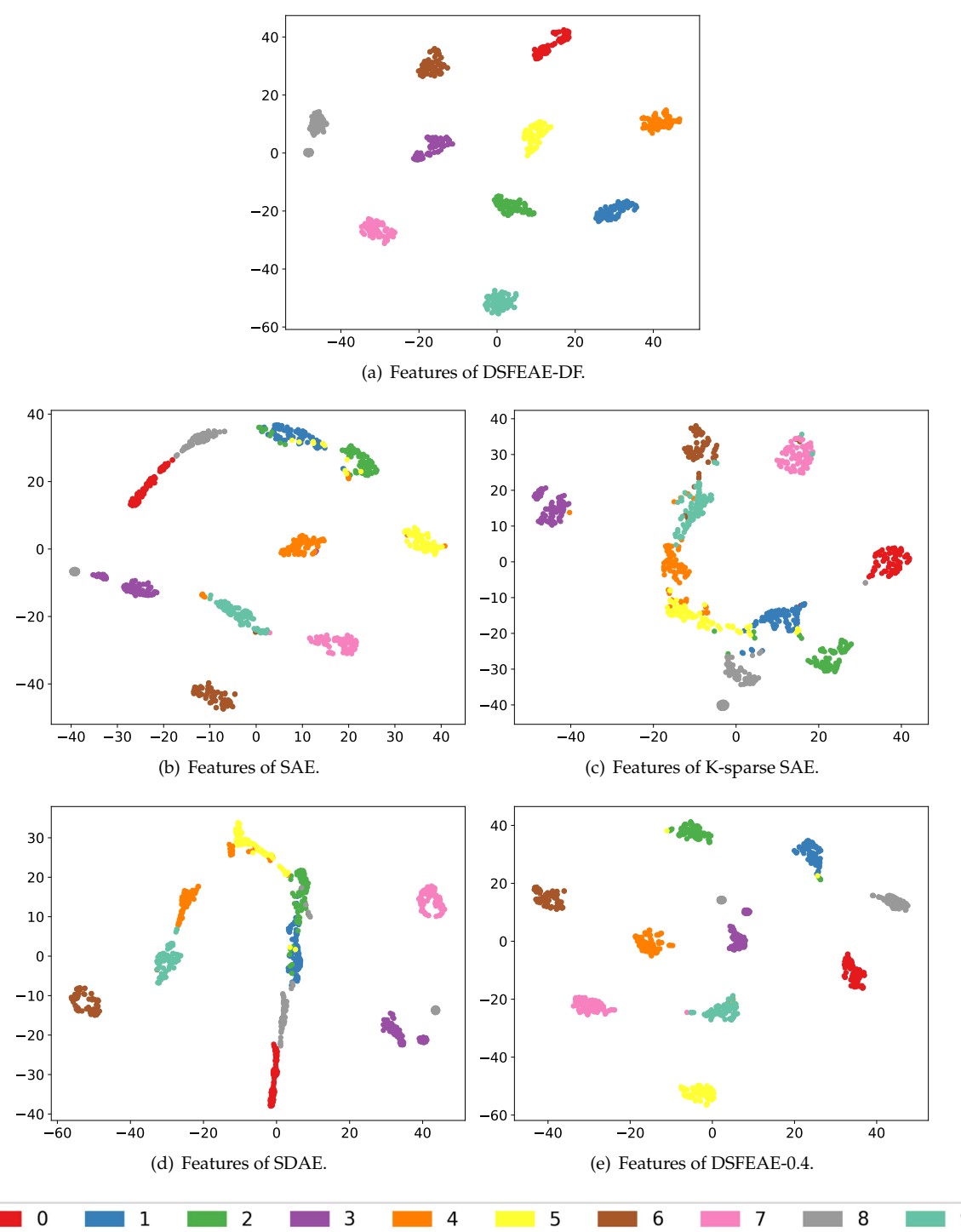

**Figure 7.** Features of five models.

## 4.5. Performance Under Noise Environment

To verify the superiority of the proposed method in real WTs, the additive white Gaussian noise is added to the testing samples to synthesize signals with different signal-to-noise ratios to simulate the actual working conditions. The definition of the signal-to-noise ratio is defined as follows:

$$\text{SNR (dB)} = 10 \log \left( \frac{\text{P}_{\text{signal}}}{\text{P}_{\text{noise}}} \right), \tag{13}$$

where $\text{P}_{\text{signal}}$ and $\text{P}_{\text{noise}}$ are the power of the original signal and noise, respectively.

Experiments of SAE, K-sparse SAE, SDAE, DSFEAE-0.4 and DSFEAE-DF under different noise environments are conducted, whose results are shown in Table 2 and Figure 8. According to Figure 8, when the SNR is 0, the DSFEAE-DF accuracy is as high as 91.55% which is much higher than the other four models. When the SNR value is less than 0, the accuracy gaps between DSFEAE-DF and the other four models are obvious. The standard deviations of DSFEAE-DF are smaller than the other four methods, which means strong stability. When the SNR is greater than 0, the accuracies of DSFEAE-DF are still much higher than the accuracies of SAE, K-sparse SAE, and SDAE. Meanwhile, the accuracy gaps between DSFEAE-DF and DSFEAE-0.4 become tiny but DSFEAE-DF still has advantages with the fact that when SNR=10, the accuracies of DSFEAE-DF and DSFEAE-0.4 are 98.20% and 98.00%, respectively. These results can verify the superiority of the proposed method under noise environment.

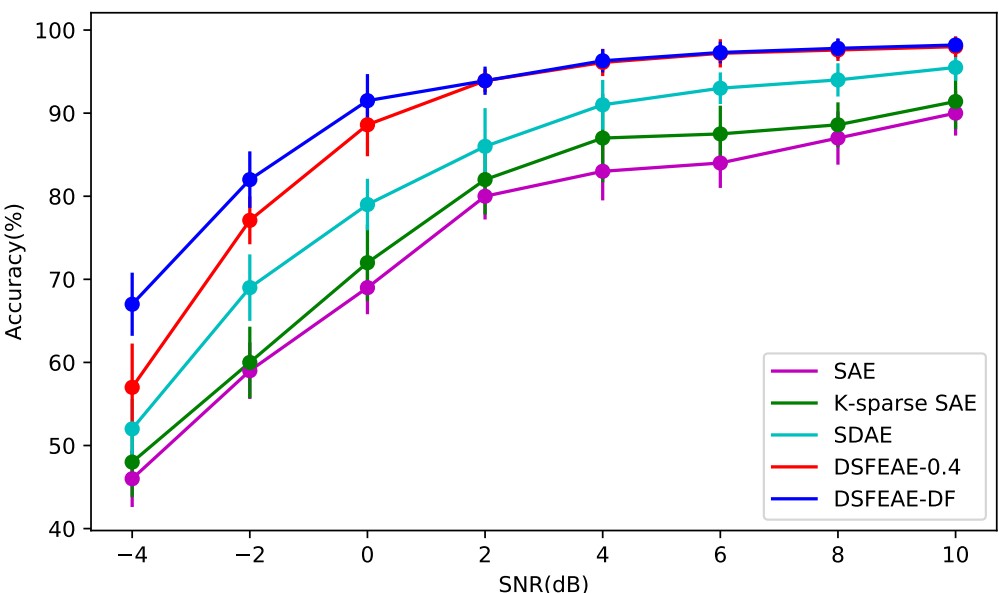

**Figure 8.** Accuracies with different SNR.

## 4.6. Visualization of Network

In order to visualize the reactions of neurons to gain some insights of feature enhancement, the features of random 200 testing samples of each hidden layer of DSFEAE-DF model and SDAE model are shown in Figure 9. In Figure 9a, the $\{f^{i,1}\}_{i=1}^{200}$, $\{f^{i,2}\}_{i=1}^{200}$ and $\{f^{i,3}\}_{i=1}^{200}$ mean the features of the *l*-th FEAE, and $\{h^{i,1}\}_{i=1}^{200}$, $\{h^{i,2}\}_{i=1}^{200}$ and $\{h^{i,3}\}_{i=1}^{200}$ represent the enhanced features of the *l*-th FEAE, respectively. In Figure 9b, the $\{f^{i,1}\}_{i=1}^{200}$, $\{f^{i,2}\}_{i=1}^{200}$ and $\{f^{i,3}\}_{i=1}^{200}$ mean the features of *l*-th DAE, respectively. In Figure 9b, a large number of features in $\{f^{i,1}\}_{i=1}^{200}$, $\{f^{i,2}\}_{i=1}^{200}$ and $\{f^{i,3}\}_{i=1}^{200}$ are activated slightly, which contains too much redundant information, increases the similarity of similar fault features, and degrades the diagnosis of similar faults seriously. While in Figure 9a, a certain amount features in $\{f^{i,1}\}_{i=1}^{200}$ and $\{f^{i,2}\}_{i=1}^{200}$ are enhanced to $\{h^{i,1}\}_{i=1}^{200}$, $\{h^{i,2}\}_{i=1}^{200}$ in the first feature enhanced layer and the second feature enhanced layer. And the rest features are suppressed, which can increase

the feature diversity between different samples. In the third FEAE, the most of the features in $\left\{ f^{i,3} \right\}_{i=1}^{200}$ are enhanced with a relatively large $\alpha^{b,3}$. That is because the features after dimension reduction continue to decrease, but sufficient features must be retained and enhanced to prevent loss of sample information. These visualizations show the insights of feature enhancement.

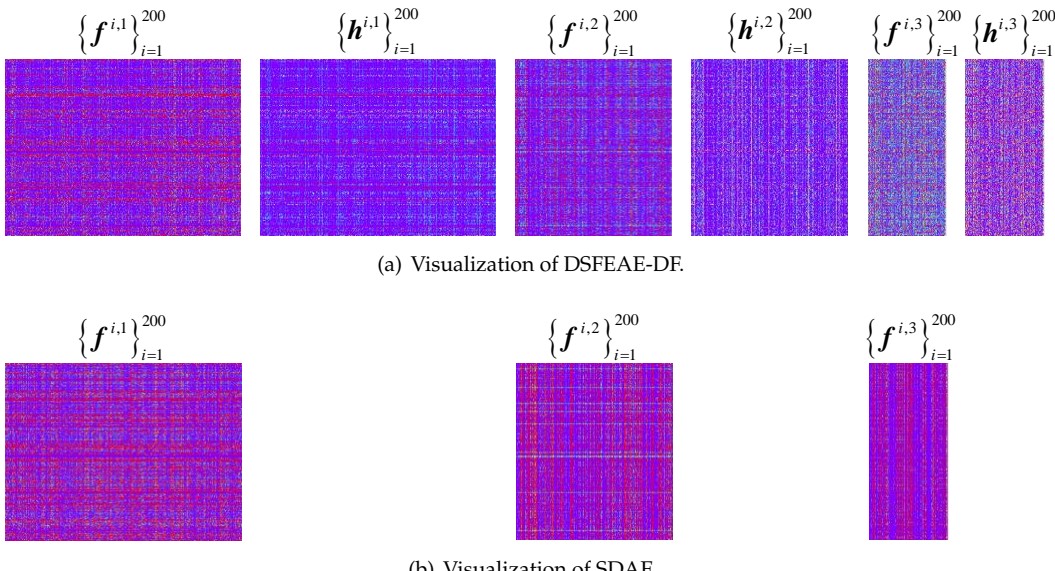

(a) Visualization of DSFEAE-DF.

(b) Visualization of SDAE

**Figure 9.** Visualization of DSFEAE-DF and stacked denoising autoencoder (SDAE)

## 5. Conclusions

In this paper, a novel model called denoising stacked feature enhanced autoencoder with dynamic feature enhanced factor (DSFEAE-DF) is proposed for fault diagnosis. The model, integrating a noise adding scheme, stacked feature enhanced autoencoders and dynamic feature enhanced factor, is proposed to discover more discriminative features from raw signals. Compared with traditional approaches, such as SAE, K-sparse SAE, SDAE and DSFEAE-0.4, our proposed method achieves superior performances in similar faults diagnosis and noise environment faults diagnosis. In addition, the reactions of neurons are visualized to show the insights of feature enhancement.

**Author Contributions:** Conceptualization, X.N. and S.L.; methodology, S.L.; software, X.N. and S.L.; validation, X.N. and S.L.; writing–original draft preparation, X.N.; writing–review and editing, X.N.; supervision, G.X.; project administration, G.X.; funding acquisition, X.N. and G.X. All authors have read and agreed to the published version of the manuscript.

**Funding:** This research was funded by the Key Research and Development Plan of Shanxi Province (Grant No. 201703D111027 ), Shanxi International Cooperation Project (Grant No. 201803D421039, Grant No. 201903D421045), and the Foundation of Shanxi Key Laboratory of Advanced Control and Equipment Intelligence (Grant No. ACEI202001).

**Conflicts of Interest:** The authors declare no conflict of interest.

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
