# Peer review of "A Novel Autoencoder with Dynamic Feature Enhanced Factor for Fault Diagnosis of Wind Turbine"

_electronics, doi:10.3390/electronics9040600_

Round 1

Reviewer 1 Report

This paper proposes a stack of feature enhanced autoencoders capable of diagnosing wind turbine faults from noisy vibrations signals. The problem and solution are well motivated and they are technically sound. However, the manuscript has important shortcomings.

Section 2 needs some improvements. When an autoencoder is introduced, class information (y_i) is not necessary, so it should be omitted. Equation (1) should be defined on two sets of data (X, Z) or ({x_i},{x_z}), not on a pair (x_i, z_i) as it is done now. The principle of operation of the autoencoder should be explained in more detail.

In section 3, although the dependence of the enhanced factor on the similarity sim and on the information amount I can be deduced, it is recommended to better explain the rationale of equation (10).

The main question about the work is the separate training of the layers. This means that layer 1 is trained to reconstruct the noisy signal of the input, is that so? Is this better than training all layers together, so that they aim to rebuild the original denoised signal? Was this empirically tested?

The database should be better explained in section 4. For example, what exactly is a sample? How are samples extracted from the signals? What is the duration of the signals?

Regarding the design of the DSFEAE, it is not explained how the values of the hyperparameters (learning rate, epochs, batch size, layer sizes, etc.) were chosen. Normally, the optimization of hyperparameters is carried out exclusively from training samples; however, nothing is said about it.

Some formal aspects should also be addressed: 1) the best result by column in table 2 can be highlighted; 2) the scales in which the accuracy is expressed in the tables (0-100) and in figure 7 (0-1) should be unified; 3) what implementations of the state of the art methods were used?

Finally, the manuscript requires a moderate linguistic revision to correct typos (e.g. 'dencoding', 'construed') and wrong structures (e.g. 'The main contributions of this paper are three fords').

Reviewer 2 Report

Improve in methodology towards validation

give more details about experimental setup. Authors should explain similar of this testing against real ones in widn turbines

conclusions needs improvements

Minor corrections

Line 76  DSFEAE-DF will be detailed (avoid using future)

figure 4 needs more explanation 

Round 2

Reviewer 1 Report

Most issues have been addressed, but some relevant ones require more attention.

The procedure to extract the segments is not sufficiently explained. How long is the signal of each fault captured? What is the relationship between a signal and its derived segments? How long is a segment? Why does a segment have 1200 points? Where does the amount of 8000 come from?

Please explain clearly how the values of the hyperparameters were chosen. Were others tried? How did they compare? How were decisions made?

There are still many linguistic errors or inaccuracies. The manuscript must be revised by a specialized service.

Round 3

Reviewer 1 Report

Although some experimental design decisions are not the most appropriate, they have been well explained in the current version of the manuscript. Finally I recommend a last revision of formal and linguistic aspects. Congratulations.
